# Modulation of Macrophage Redox and Apoptotic Processes to *Leishmania infantum* during Coinfection with the Tick-Borne Bacteria *Borrelia burgdorferi*

**DOI:** 10.3390/pathogens12091128

**Published:** 2023-09-04

**Authors:** Danielle Pessôa-Pereira, Breanna M. Scorza, Karen I. Cyndari, Erin A. Beasley, Christine A. Petersen

**Affiliations:** 1Department of Epidemiology, College of Public Health, University of Iowa, Iowa City, IA 52242, USA; dpessoapereira@uiowa.edu (D.P.-P.); breanna.scorza@gmail.com (B.M.S.); erinbeasley6@gmail.com (E.A.B.); 2Center for Emerging Infectious Diseases, University of Iowa, Iowa City, IA 52242, USA; karen-cyndari@uiowa.edu; 3Department of Emergency Medicine, University of Iowa Hospitals and Clinics, Iowa City, IA 52242, USA

**Keywords:** canine leishmaniosis, Lyme disease, coinfection, progression, inflammation, apoptosis

## Abstract

Canine leishmaniosis (CanL) is a zoonotic disease caused by protozoan *Leishmania infantum*. Dogs with CanL are often coinfected with tick-borne bacterial pathogens, including *Borrelia burgdorferi* in the United States. These coinfections have been causally associated with hastened disease progression and mortality. However, the specific cellular mechanisms of how coinfections affect microbicidal responses against *L. infantum* are unknown. We hypothesized that *B. burgdorferi* coinfection impacts host macrophage effector functions, prompting *L. infantum* intracellular survival. In vitro experiments demonstrated that exposure to *B. burgdorferi* spirochetes significantly increased *L. infantum* parasite burden and pro-inflammatory responses in DH82 canine macrophage cells. Induction of cell death and generation of mitochondrial ROS were significantly decreased in coinfected DH82 cells compared to uninfected and *L. infantum*-infected cells. Ex vivo stimulation of PBMCs from *L. infantum*-seronegative and -seropositive subclinical dogs with spirochetes and/or total *Leishmania* antigens promoted limited induction of IFNγ. Coexposure significantly induced expression of pro-inflammatory cytokines and chemokines associated with Th17 differentiation and neutrophilic and monocytic recruitment in PBMCs from *L. infantum*-seropositive dogs. Excessive pro-inflammatory responses have previously been shown to cause CanL pathology. This work supports effective tick prevention and risk management of coinfections as critical strategies to prevent and control *L. infantum* progression in dogs.

## 1. Introduction

Visceral Leishmaniasis (VL) is a vector-borne zoonotic disease caused *by Leishmania infantum*, an obligate intracellular protozoan parasite [1]. Macrophages are the predominant host cells for *Leishmania*, generating reactive oxygen species (ROS) to kill intracellular parasites and limit their replication. Macrophages also produce pro-inflammatory cytokines to induce CD4 + T cell activation that modulate immune responses against *Leishmania* [2]. Th1 cell responses have been shown to be crucial for macrophage activation and subsequent control of parasite burden [3].

Dogs are reservoir hosts for *L. infantum* and play a central role in domestic transmission to humans [4,5]. Not only are polysymptomatic and highly parasitemic dogs associated with increased infectiousness to sand fly vectors [6,7], but multiple studies indicate that dogs with subclinical or limited canine leishmaniosis (CanL) also readily transmit parasites to sand flies [8,9]. Currently, *L. infantum* infection of U.S. dogs is predominantly maintained through vertical transmission [10,11]. Although vector transmission has not yet been reported within the U.S., concerns about climate change impacting the geographic distribution of sand fly vectors already found in North America are mounting [12,13,14,15]. As higher prevalence of competent vectors increases the risk of human exposure, surveillance of *L. infantum* infection among dogs, and understanding the risk factors associated with infection and transmission among dogs, will be critical to preventing VL from becoming endemic in the U.S.

Recent epidemiological findings implicate comorbid tick-borne coinfections with progression to clinical CanL [16,17,18]. Dogs exposed to tick-borne pathogens are significantly more likely to be seropositive to *L. infantum* in Brazil and the U.S. Furthermore, exposure to tick-borne bacteria was significantly associated with CanL progression and mortality [18]. Within the U.S., Lyme Disease (LD), caused by the spirochete *Borrelia burgdorferi*, is one of the most common tick-borne diseases found in hunting and sporting dogs (39% seropositivity) [19], including those with CanL [18]. *Leishmania* and *Borrelia* spp. coinfections have been found in dogs in Europe, where zoonotic VL transmission also occurs via sand fly vectors [20].

*B. burgdorferi* infects a wide range of host cell types, including important host cells for *Leishmania*, such as macrophages [21]. *Leishmania* and *Borrelia* spp. have evolved evasion mechanisms to modulate host cell immune responses and resist destruction [22,23]. Following sand fly inoculation, *Leishmania* promastigotes are initially phagocytosed by neutrophils, which fail to support infection and undergo apoptosis [24,25,26]. Typically, apoptosis is a mechanism employed by host immune cells to eliminate intracellular pathogens. Instead, this process enables *Leishmania* to silently enter macrophages by way of apoptotic bodies containing viable *Leishmania* [24,25,26]. Within macrophages, *Leishmania* confer resistance to apoptosis to secure intracellular survival [27,28,29,30]. *Leishmania donovani*, another visceralizing species, inhibits pro-apoptotic signaling of p38 mitogen-activated protein kinase (MAPK) in human macrophages [31,32]. *Leishmania* upregulation of macrophage Bcl-2 and activation of Akt decreases cell death rates [33]. Studies of *B. burgdorferi* regulating host cell apoptosis are conflicting [34,35]; thus, it is uncertain how *L. infantum* and *B. burgdorferi* coinfection would affect host cell survival.

Macrophages utilize ROS to kill phagocytosed pathogens [2,36]. *Leishmania* can inhibit Nox2 maturation and assembly of NADPH oxidase at the phagolysosomal membrane, thus preventing ROS generation [37,38]. *B. burgdorferi* is resistant to direct ROS-mediated damage [39,40,41]. *B. burgdorferi* prevents peripheral blood mononuclear cell (PBMC) NADPH-dependent ROS generation in response to secondary stimuli through upregulation of the Akt-mTOR pathway [42,43]. This is a mechanism that could possibly favor susceptibility to coinfection with *Leishmania*. Another source of ROS in macrophages is the mitochondria [36]. The role of mtROS in defense against *Leishmania* is relatively unexplored. mtROS contributed to increased clearance of intracellular *L. donovani* [44,45] but mtROS in *L. infantum* clearance remains largely unknown.

While Type 1 immunity is crucial for controlling *Leishmania* infection, Type 2 and Type 17 responses are thought to be detrimental by skewing the immune response [3,46]. Type 17 immune responses occur during *Borrelia* spp. infections [47,48]. Th17 cell differentiation is induced by IL-23, IL-6, IL-1β, and TGF-β, resulting in the production of highly inflammatory Type 17 effector cytokines such as IL-17 and IL-22 [49]. Although *L. infantum* infection in dogs upregulates *IL17A* gene expression [50,51], sustained production of IL-17A increases neutrophil recruitment to inflammatory sites, contributing to immunopathology [46]. We speculate *L. infantum* and *B. burgdorferi*-coinfected dogs would display mixed Th1/Th17 responses, which could undermine *L. infantum* infection control and clinical outcomes.

We hypothesize *B. burgdorferi* coinfection negatively affects control of *L. infantum* infection by adopting evasion mechanisms that alter macrophage effector functions. Here, we demonstrate that in vitro coexposure to *L. infantum* and *B. burgdorferi* promotes robust alterations in canine macrophage inflammatory responses, cell death, and mitochondrial ROS production, contributing to enhanced *L. infantum* survival. Additionally, exposing *L. infantum*-seropositive dog PBMCs to *B. burgdorferi* induced a Type 17-like pro-inflammatory response. Together, our data suggests *B. burgdorferi* coinfection drives a skewed inflammatory response that we found associated with clinical coinfection.

## 2. Materials and Methods

### 2.1. Cell Culture

Canine monocyte-macrophage (DH82) (cat# CRL-10389) and murine macrophage RAW 264.7 (cat# TIB-71) cell lines were originally obtained from the American Type Culture Collection (ATCC, Manassas, VA, USA). DH82 and RAW 264.7 cells were maintained in Dulbecco’s modified Eagle’s medium (DMEM 1X, 4.5 g/L glucose and L-glutamine; Gibco, Waltham, MA, USA) supplemented with either 10% or 1% heat-inactivated fetal bovine serum (FBS) (R&D Systems, Minneapolis, MI, USA), 1% MEM non-essential amino acids solution (100×) (Gibco, Waltham, MA, USA), and 1% penicillin/streptomycin solution (P/S) (Gibco, Waltham, MA, USA), at 37 °C and 5% CO_2_. Adherent cells were harvested using 0.25% trypsin-EDTA solution (Gibco, Waltham, MA, USA). Seventy-two hours prior to infection, cell culture media was replaced with fresh media without antibiotics.

### 2.2. Animals

Fourteen naturally *L. infantum*-exposed, US-owned foxhounds were identified via *Leishmania* serology using the Dual Path Platform (DPP)^®^ CVL assay (Bio-Manguinhos, Manguinhos, Rio de Janeiro, Brazil). All dogs were seronegative for *Anaplasma phagocytophilum*, *Anaplasma platys*, *Ehrlichia ewingii*, *Ehrlichia canis*, and *B. burgdorferi* according to the IDEXX 4Dx SNAP test (IDEXX Reference Labs Inc., Westbrook, ME, USA). Complete physical examinations were performed by licensed veterinarians and peripheral whole blood and serum samples were collected for complete blood count, serum chemistry, *L. infantum* detection via quantitative PCR (qPCR), as previously described [52], and canine peripheral blood mononuclear cells (PBMC) isolation. CanL clinical staging was determined according to the LeishVet scoring system [53]. Dogs showing severe, disseminated disease were excluded (LeishVet score 4 and qPCR positive dogs). The demographics and clinical parameters are described in Table 1. All animal work was reviewed and approved by the University of Iowa Institutional Animal Care and Use Committee (IACUC) (Appendix A).

### 2.3. Parasites and Bacteria

*L. infantum* parasites (US/2016/MON1/FOXYMO4), originally isolated from a naturally infected U.S. dog, were cultured in complete hemoflagellate-modified minimal essential medium (HOMEM) supplemented with 10% FBS (R&D Systems, Minneapolis, MI, USA) at 26 °C. For infections, stationary-phase promastigotes were counted on a hematocytometer using a light microscope (Olympus CKX41, Olympus Corporation, Tokyo, Japan) at 400× magnification, washed, and resuspended in DMEM or RPMI (RPMI-1640 1X, L-glutamine; Gibco, Waltham, MA, USA) without P/S at 1:10 MOI (multiplicity of infection; cell:pathogen ratio).

Low passage isolates of *B. burgdorferi* spirochetes strain B31; clone 5A1; NR-13251 (BEI Resources Repository, USA) were maintained in complete Barbour–Stonner–Kelly (BSK-H; Sigma-Aldrich, Burlington, MA, USA) medium with 6% rabbit serum at 34 °C. For infections, late-exponential phase spirochetes were counted via phase-contrast microscopy (Olympus BX41, Olympus Corporation, Tokyo, Japan) at 400× magnification, washed, and resuspended at either 1:10 or 1:25 MOI.

### 2.4. PBMC Stimulation Assay

Diluted whole-blood samples were underlaid with Ficoll–Paque Plus (Cytiva, Marlborough, MA, USA) and then centrifugated at 500× *g* for 30 min at 20 °C without brakes. Mononuclear cell layers were collected, washed in Dulbecco’s phosphate-buffered saline (DPBS 1X, Gibco, Waltham, MA, USA), and plated at 5 × 10^5^ cells/well into 96-well round bottom plates (Corning Costar, Waltham, MA, USA). For the qPCR assays, PBMCs from each dog were stimulated with either vehicle (RPMI; 10% heat inactivated FBS; Gibco, Waltham, MA, USA) or live *B. burgdorferi* spirochetes at 1:10 MOI for 24 h at 37 °C and 5% CO_2_. For the ELISAs, PBMCs from each dog were stimulated with either vehicle, 10 µg/mL total *Leishmania* antigen (TLA), live *B. burgdorferi* spirochetes at 1:10 MOI, or TLA and *B. burgdorferi* combined for 72 h at 37 °C and 5% CO^2^.

### 2.5. Assessment of Leishmania In Vitro Burden

DH82 (2 × 10^5^/well) and RAW 264.7 (5 × 10^4^/well) cells were seeded in 24-well tissue culture-treated plates containing sterilized 12 mm glass coverslips (Fisherbrand, Waltham, MA, USA) and incubated at 37 °C, 5% CO_2_ overnight, as described elsewhere [54]. The following day, cells were infected with *L. infantum* at 1:10 MOI. Two hours post-infection, wells were thoroughly washed with pre-warmed supplemented DMEM to remove non-internalized parasites and *B. burgdorferi* were inoculated at 1:25 MOI. 24 h and 48 h post coinfections, coverslips were fixed with methanol (Research Products International, Mount Prospect, IL, USA), stained with HEMA 3 (Thermo Scientific, Waltham, MA, USA) according to the manufacturer’s instructions. Slides were assessed under a light microscope (Olympus BX41, Olympus Corporation, Tokyo, Japan) at 1000× magnification by blinded reviewers to quantify percent *L. infantum*-infected cells and number of intracellular amastigotes per 100 cells.

### 2.6. Apoptosis Assay

DH82 cells were seeded in 24-well plates and infected with *L. infantum* and/or *B. burgdorferi* as previously described. After 24 h post coinfections, cells were harvested using 0.25% trypsin-EDTA and allowed to recover for 30 min in supplemented DMEM at 37 °C, 5% CO_2_ before resuspension and labeling with Annexin V-A488 and Propidium iodide (PI) using the Dead Cell Apoptosis Kit (cat# V13241; Invitrogen, Waltham, MA, USA), according to the manufacturer’s protocol. Cell suspensions underwent flow cytometry (Becton Dickinson LSR II flow cytometer; BD Biosciences, Franklin Lakes, NJ, USA) at the University of Iowa Flow Cytometry Core Facility counting at least 10.000 events/test. Data were analyzed using FlowJo V10.8 software. The positive control for cell death was uninfected DH82 cells treated with 10 µM camptothecin (CPT; Thermo Fisher Scientific, Waltham, MA, USA).

### 2.7. Reactive Oxygen Species (ROS) Detection

CellROX Green and MitoSOX Red (Thermo Fisher, Waltham, MA, USA) were used to determine total and mitochondrial ROS production, respectively, in live cells through flow cytometry. Briefly, DH82 cells (1 × 10^5^/well) were seeded in 24-well plates and infected with *L. infantum* and/or *B. burgdorferi* in DMEM supplemented with 1% heat inactivated FBS. For measuring total ROS, cells were incorporated with 1 µM CellROX Green reagent for 30 min 2 h post-coinfections. Cells were then harvested, washed, and resuspended in DPBS before undergoing flow cytometry. For measuring mitochondrial ROS (mtROS), cells were incorporated with 2.5 µM MitoSOX Red reagent for 30 min 1 h post-coinfections. Cells were harvested, washed, and resuspended in HBSS with ions (Gibco, USA) before undergoing flow cytometry. Positive controls for total ROS and mtROS generation included uninfected DH82 cells treated with 1 µg/mL lipopolysaccharide (LPS) from *Escherichia coli* 0111:B4 (Sigma-Aldrich, Burlington, MA, USA) or 20 µM rotenone (Sigma-Aldrich, Burlington, MA, USA), respectively. Flow cytometry was performed using the Becton Dickinson LSR II flow cytometer (BD Biosciences, Franklin Lakes, NJ, USA) at the University of Iowa Flow Cytometry Core Facility. Measurement of fluorescence in cells was made by counting at least 10.000 events/test. Data were analyzed using FlowJo V10.8 software.

### 2.8. RNA Isolation and Quantitative PCR

After infection and/or treatment, plated DH82 cells and PBMCs were harvested and resuspended in Trizol (Invitrogen, Waltham, MA, USA) for total RNA isolation according to the manufacturer’s instructions. Quantity and quality assessment of RNA samples were measured by NanoDrop 2000 (Thermo Fisher Scientific, Waltham, MA, USA). Total RNA samples from DH82 cells (250 ng) and PBMCs (125 ng) were used for cDNA synthesis using the SuperScript™ III First-Strand Synthesis System (Thermo Fisher Scientific, Waltham, MA, USA), according to the manufacturer’s instructions. After reverse transcriptions, cDNA samples were stored at −20 °C until further analysis.

For gene expression analyses, real-time qPCR (RT-qPCR) was performed using Power SYBR Master Mix (Thermo Fisher Scientific, Waltham, MA, USA), on a QuantStudio 3 Real-Time PCR System thermocycler (Thermo Fisher Scientific, Waltham, MA, USA). Each reaction was performed in a final volume of 10 µL containing 2 µL of diluted cDNA (1:5), 0.5 µM of each primer, and 1X Power SYBR Master Mix. Cycling conditions were as follows: 2 min at 55 °C, 2 min at 95 °C, followed by 40 cycles of 15 s at 95 °C, and 1 min at 60 °C. At the end of the PCR cycles, melting curve analyses were performed in order to validate the specific amplification of the target transcript. Each sample was run in triplicate, and expression levels were normalized to *B2M* and/or *ACTB*, when indicated. Log2-fold changes (log2FC) were calculated using the 2^−ΔΔCT^ method, with uninfected DH82 or unstimulated PBMCs from *L. infantum*-seronegative dogs used as control groups. Target-specific forward and reverse primers are described in Appendix A.

### 2.9. Cytokine BioPlex and ELISA

Supernatant cytokines and chemokines were assayed using a Milliplex Canine Cytokine/Chemokine Magnetic Bead Panel kit (custom premixed 7-plex; Millipore Sigma, Burlington, MA, USA) according to the manufacturer instructions, which allowed the simultaneous quantification of IFN-γ; IL-6; TNF-α; IL-10; IL-8; KC-like (CXCL-1); and MCP-1 (CCL2). Plates were read on the Bio-Plex Luminex 2000 (Bio-Rad, Hercules, CA, USA) at the University of Iowa Flow Cytometry Core Facility. Supernatant IL-17A was quantified using the Canine IL-17A DuoSet ELISA kit (R&D Systems, Minneapolis, MI, USA), according to the manufacturer instructions. Plates were read on VersaMax microplate reader (Molecular Devices, San Jose, CA, USA).

### 2.10. Statistical Analyses

Statistical analyses were performed using GraphPad Prism (v. 9.4.1, USA). Prior to analysis, outliers were removed using a ROUT test (Q = 1%). An ordinary one-way analysis of variance (ANOVA) with Tukey’s post hoc test was performed for multiple comparisons between groups within the same timepoint. Two-way ANOVA with either Tukey’s or Sidak’s post hoc tests were performed for comparisons between groups within different timepoints, as appropriate. For non-parametric tests, either Kruskal–Wallis with Dunn’s multiple comparisons or Wilcoxon matched-pairs rank test with Holm–Sidak post-test were performed. Error bars denote mean and either standard deviation (SD) or standard error of the mean (SEM) as indicated.

## 3. Results

### 3.1. B. burgdorferi Coinfection Promotes L. infantum Infection in Macrophages

To understand how tick-borne pathogens alter the outcome of intracellular *L. infantum* infection, we coinfected *L. infantum* and *B. burgdorferi* in host macrophages. The initial question was whether *B. burgdorferi* modulated *L. infantum* infection of canine macrophage DH82 cells. We found that DH82 macrophages maintained a steady parasite infection rate and burden over 48 h in culture (Figure 1). However, *B. burgdorferi*-coinfected macrophages showed significantly higher *Leishmania* infection frequencies compared to *L. infantum*-infected cells at both 24 h (21.8% increase; *p* = 0.0001) and 48 h (31.4% increase; *p* = 0.0001) post *B. burgdorferi* addition (Figure 1A). Numbers of intracellular parasites were also significantly higher in *B. burgdorferi* coinfected macrophages than *L. infantum*-infected cells at 24 h (25.6% increase; *p* = 0.0001) and 48 h (68.1% increase; *p* < 0.0001) post *B. burgdorferi* addition (Figure 1B). We additionally measured the difference in parasite infection and burden rates between *Leishmania* infection alone or *B. burgdorferi* coinfection in RAW 264.7 murine macrophage cell line (Appendix A). RAW 264.7 macrophages showed a signification reduction in parasite infection and burden at 24 h and 48 h compared to the initial infection (*p* < 0.0001). However, there was not a significant reduction in parasite infection and burden at 24 h or 48 h compared to the initial infection among *B. burgdorferi* coinfected RAW 264.7 cells. Further, at 24 h and 48 h time points, the coinfected group showed a significantly higher rate of *Leishmania* infection and burden compared to the *Leishmania*-infected macrophages alone (*p* < 0.0001). Taken together, *B. burgdorferi* coinfection contributed to significantly increased frequency of infection and parasite load in *L. infantum*-infected macrophages in vitro.

### 3.2. B. burgdorferi Coinfection Decreases Cell Death in L. infantum-Infected Macrophages

To explore the possible mechanism by which *B. burgdorferi* infections leads to increased parasite load in DH82 cells, we investigated the rate of apoptosis and necrosis of *L. infantum*-infected DH82 cells in the presence or absence of *B. burgdorferi* spirochetes via flow cytometry using Annexin V and PI staining (gating strategy shown in Appendix A). Treatment with Camptothecin (CPT), a topoisomerase I inhibitor, was used as positive control for cell death (*p* < 0.0001; as compared to all groups) (Figure 2A). The frequency of *L. infantum*-infected cells undergoing cell death (Annexin V+) did not significantly differ from uninfected cells (19.1% vs. 22.8%, respectively) over 24 h in culture. On the other hand, cells coinfected with *L. infantum* and *B. burgdorferi* for 24 h showed significantly lower death rates compared to uninfected cells (40.1% decrease; *p* < 0.0001) and *L. infantum*-infected cells (28.4% decrease; *p* = 0.04).

By discriminating early apoptotic (Annexin V+/PI−) from late apoptotic/necrotic cells (Annexin V+/PI+), we could explore which category of cell death was most affected by *B. burgdorferi* coinfection. While early apoptosis was not significantly altered by any experimental group, reduced late apoptosis/necrosis was observed in coinfected cells compared to uninfected cells (~41% decrease; *p* < 0.0001) and *L. infantum*-infected cells (~28% decrease; *p* = 0.01) (Figure 2C). These findings showed that *B. burgdorferi* coinfection significantly limits apoptosis in *L. infantum*-infected DH82 macrophages.

### 3.3. B. burgdorferi Coinfection Suppresses Mitochondrial ROS Production in L. infantum-Infected Macrophages

Macrophages produce ROS in response to pathogen- and damage-associated molecular patterns (PAMPs and DAMPs, respectively) signals to kill intracellular or phagocytosed pathogens. It has been shown that *B. burgdorferi* can suppress ROS generation of host cells [43]. We hypothesize that this evasion mechanism occurs in *L. infantum*-infected macrophages when coexposed to *B. burgdorferi*. Therefore, we performed qPCR to explore relative transcript levels of genes involved in ROS pathways and measured ROS levels in *L. infantum*-infected DH82 cells before and after *B. burgdorferi* coinfection (Figure 3).

NOX2 expression, a NADPH oxidase subunit, remained relatively stable in *L. infantum*-infected DH82 cells. However, 24 h after *B. burgdorferi* addition, we found significant downregulation of NOX2 transcript in coinfected macrophages compared *L. infantum*-infected cells (*p* = 0.002) (Figure 3A).

We next tested whether *B. burgdorferi* exposure resulted in decreased total ROS generation in *L. infantum*-infected DH82 cells using CellROX-Green followed by flow cytometry analysis (gating strategy shown in Appendix A). Treatment with LPS was used as a positive control for oxidative stress (*p* < 0.0001; compared to unstimulated cells) (Figure 3B). *L. infantum*-infected macrophages generated robust levels of total ROS, displaying significantly increased %CellROX^HI^ cells compared to uninfected cells (50.96% increase, *p* = 0.0071). *B. burgdorferi* addition alone did not lead to total ROS generation as %CellROX^HI^ cells were comparable to uninfected cells. However, when added to cells already infected with *L. infantum*, *B. burgdorferi* did not inhibit the total ROS generated in response to the *L. infantum*, as the %CellROX^HI^ cells were not significantly different in coinfected cells compared to *L. infantum*-infected cells 2 h post-coinfection. Similar patterns were observed using the mean fluorescence intensity (MFI) of CellROX (Appendix A), suggesting that *L. infantum*-infected DH82 cells display enhanced total ROS production, which is not affected after coinfection with *B. burgdorferi*.

To better identify the intracellular source of ROS production, we assessed if coinfection was specifically altering mtROS production. We assayed superoxide dismutase 2 (SOD2) transcripts, an antioxidant mitochondrial enzyme known to neutralize superoxide and limit damage from ROS [55]. There was significantly increased SOD2 gene expression in coinfected cells at 2 h (*p* = 0.0006) and 24 h (*p* = 0.0114) post-coinfection compared to *L. infantum*-infected cells (Figure 3D). We measured mtROS production using MitoSOX fluorescent labeling and flow cytometry (gating strategy shown in Appendix A). MitoSOX accumulates in the mitochondrial matrix and specifically binds superoxide anions (36). Similar to what we observed with total ROS, *L. infantum*-infected DH82 cells presented significantly higher %MitoSOX^HI^ compared to uninfected cells (140.2% increase, *p* < 0.0001) (Figure 3E). However, unlike with total ROS, after *B. burgdorferi* coinfection the proportion of MitoSOXHI cells was significantly decreased compared to cells infected with *L. infantum* alone (57.49% decrease, *p* < 0.0001) (Figure 3F). *B. burgdorferi* exposure also resulted in decreased MitoSOX^HI^ MFI within *L. infantum*-coinfected macrophages compared to cells exposed to *L. infantum* (1.4- vs. 2.5- fold decrease, respectively; *p* ≤ 0.0001) (Appendix A), demonstrating that *B. burgdorferi* dampens mitochondrial superoxide generation in coinfected cells, possibly through upregulation of host SOD2.

### 3.4. B. burgdorferi Infection Induces Transcription and Secretion of Pro-Inflammatory Cytokines in L. infantum-Infected Macrophages

Macrophages produce high levels of pro-inflammatory cytokines upon exposure to *B. burgdorferi* [34,56]. However, *L. infantum* is known to employ multiple mechanisms to dampen pro-inflammatory responses including upregulating protein tyrosine phosphatase activity and lipid raft disruption to inhibit JAK/STAT, PKC, NFkB, and MAPK signaling cascades downstream of pathogen recognition receptors (PRRs), such as TLRs and cytokine receptors such as IFNγR [57,58,59]. Thus, we hypothesized that *B. burgdorferi*-induced macrophage cytokine production would be attenuated in *L. infantum*-infected DH82 cells. RT-qPCR was performed to measure cytokine transcript levels in DH82 cells before and after coinfection (Figure 4).

As expected, *L. infantum*-infected macrophages did not increase inflammatory cytokine transcription compared to uninfected cells (Figure 4A). In contrast, coinfection with *B. burgdorferi* led to significant upregulation of *TNFA*, *IL6*, and *IL1B* gene expression, peaking 2 h after coinfection and persisting for at least 24 h. *IL10* gene expression also showed a significant, but transient, increase at 2 h and 4 h post-coinfection (Figure 4A). Cell cultures infected with *L. infantum* prior to *B. burgdorferi* exposure showed the same pattern of gene expression as *B. burgdorferi*-infected cells. Production and release of TNF-α, IL-6, and IL-10 protein 24 h after *B. burgdorferi* infection and coinfection was confirmed via ELISA (Figure 4B). *B. burgdorferi* was capable of triggering a strong pro-inflammatory response in *L. infantum*-infected DH82 macrophages, significantly altering the microenvironment surrounding *Leishmania* parasitized host cells.

### 3.5. B. burgdorferi Induces Innate Pro-Inflammatory Responses in PBMCs from Dogs with Subclinical or Mild Canl

Clinical control of VL depends on Type I immune responses characterized by IFNγ production, primarily by activated CD4+ T cells [3,60]. IFNγ is a major activator of macrophages, sustaining microbicidal responses and enhancing antigen presentation [61,62]. As *B. burgdorferi* infection promoted significant pro-inflammatory responses in *L. infantum*-infected DH82 macrophages, we next evaluated whether experimental *B. burgdorferi* exposure altered inflammatory or regulatory cytokine production from primary dog immune cells. We performed ex vivo stimulation assays with either vehicle (media), live *B. burgdorferi* spirochetes (1:10 MOI) live *B. burgdorferi* spirochetes (1:10 MOI) comparing PBMC responses from six dogs with (DPP+) and five without (DPP-) *L. infantum* seroreactivity and clinical evidence of *L. infantum* infection. Fold changes are normalized to unstimulated *L. infantum*-seronegative dog cell transcript levels (Figure 5A).

After 24 h of *B. burgdorferi* stimulation, PBMCs from both *L. infantum* seronegative and seropositive dogs showed increased expression of *TNFA*, *IL6*, and *IL1B*, and a relatively modest upregulation of *IL10* compared to unstimulated PBMCs (Figure 5A). However, PBMCs from *L. infantum*-seropositive dogs showed a slightly higher gene induction after *B. burgdorferi* exposure, with significantly increased *TNFA*, *IL6*, *IL1B*, *IL10,* and *IL12B* (all *p* < 0.01) compared to unstimulated *L. infantum*-seropositive dog PBMCs. *B. burgdorferi* exposure did not induce differential expression of *IFNG* and *IL12A* in PBMCs from either *L. infantum*-seronegative or positive dogs (Figure 5A).

To confirm secretion of upregulated cytokine transcripts, we performed an ex-vivo stimulation assay and stimulated PBMCs from 6 DPP- and 8 DPP+ dogs with *B. burgdorferi* spirochetes, TLA, or both for 72 h and performed ELISA on culture supernatants. We found that *B. burgdorferi*-exposed PBMCs produced significantly high levels of TNFα, IL-6, and IL-10, but limited IFNγ (Figure 5B). PBMCs *from L. infantum*-seropositive dogs produced significantly more IL-10 when stimulated with both spirochetes and TLA compared to *L. infantum*-seropositive dog PBMCs stimulated with *B. burgdorferi* alone (*p* < 0.03125). *B. burgdorferi* spirochete exposure induces a strong, mixed pro- (TNFα, IL-6) and anti-inflammatory (IL-10) cytokine response from *Leishmania*-seropositive dog PBMCs. Thus, similar to what was observed by coinfection in the macrophage cell line, the inflammatory environment surrounding PBMCs from *L. infantum*-infected dogs is highly skewed by the presence of the *B. burgdorferi*.

### 3.6. B. burgdorferi Induces Type 17 Cytokine Expression in PBMCs from L. infantum-Seropositive Dogs

We observed an increase in *IL12B* transcripts, which encodes IL-12p40 subunit, within *L. infantum*-seropositive PBMCs after *B. burgdorferi* exposure. The p40 subunit of IL-12 can also dimerize with the p19 subunit of IL-23 resulting in the formation of IL-23 [63], a critical inducer of Th17 cell differentiation and maintenance [48]. Because *B. burgdorferi* is known to induce Type 17 responses, we investigated whether other Type 17 cytokines were induced in our ex vivo coexposure model.

PBMCs from *L. infantum*-seronegative and seropositive dogs were stimulated with *B. burgdorferi* spirochetes and/or TLA and gene expression analyses and ELISA were performed (Figure 6). There was upregulation of *IL23p19*, *IL17A*, and *IL22* transcripts in *B. burgdorferi*-exposed PBMCs from both *L. infantum*-seronegative and seropositive dogs compared to unstimulated cells (Figure 6A,B). We did not observe differential gene expression of *TGFB* (Figure 6A).

IL-17A, the main effector cytokine of Th17 cells, mediates production of pro-inflammatory cytokines and recruitment of neutrophils and monocytes [64]. Although not achieving statistical significance (*p* = 0.06), we observed higher levels of IL-17A production from PBMCs exposed to both *B. burgdorferi* and TLA than by cells stimulated with either pathogen alone (Figure 6). We additionally found increased levels of IL-8 (*p* = 0.015) and CXCL1 (*p* = 0.03125), potent neutrophil chemoattractants [65], after exposure to both *B. burgdorferi* spirochetes and TLA compared to PBMCs exposed to TLA alone. These findings suggest the presence of tick-borne coinfection with *B. burgdorferi* spirochetes may skew the inflammatory milieu toward a Type 17 environment in dogs with CanL.

## 4. Discussion

Because *B. burgdorferi* and *L. infantum* coinfections can occur in dogs with CanL and have overlapping cellular tropism, we believe that *B. burgdorferi* coinfections disrupt the immune balance established during subclinical CanL and contribute to development of clinical disease. Herein, we developed an in vitro *L. infantum* and *B. burgdorferi* coinfection model using a canine macrophage cell line, DH82 cells, to evaluate the impact of these coinfections. We found that addition of *B. burgdorferi* led to increased intracellular parasite infection levels and burdens within DH82 macrophages. This phenotype was reproducible in a murine macrophage cell line. This suggests that *B. burgdorferi* alters the macrophage phenotype to be more permissive to *L. infantum* replication.

Programmed cell death, or apoptosis, is believed to benefit host cells by limiting pathogen replication. *Leishmania* spp. and *B. burgdorferi* both can suppress apoptosis of infected cells as an evasion mechanism [30,34,66], so we hypothesized that the two pathogens may synergistically decrease apoptosis, consequently favoring *L. infantum* maintenance within coinfected cells. In our model, *L. infantum* infection did not impact host cell death as compared to uninfected cells. We observed *B. burgdorferi* spirochetes reduced the proportion of DH82 cells undergoing apoptosis, particularly inhibiting late apoptosis/necrosis. Resistance to apoptosis was also described in other ex vivo and in vitro models, where cells remained viable after long-term exposure to either live *B. burgdorferi* spirochetes or their virulent components (i.e., heat-killed/inactivated spirochetes and outer surface proteins) [35,67,68,69]. As we found reduced numbers of coinfected DH82 cells undergoing cell death compared to cells exposed to *L. infantum* promastigotes, it is possible *B. burgdorferi* activates anti-apoptotic/necrotic pathways that prevent cell death and facilitates *L. infantum* intracellular survival, with no synergistic effect. Further studies are necessary to determine a causal association between decreased cell death and *L. infantum* replication or maintenance after *B. burgdorferi* exposure, and the underlying molecular mechanisms.

Effective intracellular killing of *Leishmania* by macrophages requires production of ROS [2]. *NOX2* gene expression in DH82 cells remained relatively unaffected by *L. infantum* infection (Figure 3A). However, after *B. burgdorferi* exposure, a transient *NOX2* upregulation, followed by a steady downregulation trend that reached significance after 24 h was observed. This finding suggests that *L. infantum*-infected macrophages exposed to *B. burgdorferi* might produce less superoxide. *B. burgdorferi*-exposure alone did not increase the total ROS production compared to uninfected cells at the measured timepoint—a possible active evasion mechanism. However, *B. burgdorferi* was unable to reduce the total ROS generated by *L. infantum* infected cells in this model. In agreement, Kersholtz et al. reported *B. burgdorferi* exposure does not induce significant levels of total ROS in human PBMCs [43]. Other studies demonstrated *B. burgdorferi* decreases the ability of PBMCs to generate total ROS in response to a secondary stimulus through mTOR pathway activation [42,43]. Therefore, *B. burgdorferi* coinfection could potentially enhance canine host cell susceptibility to *Leishmania* by reducing ROS, possibly through synergistic activation of the mTOR pathway.

Regulated production of mtROS is crucial in eliciting effective immune responses without causing mitochondrial damage [36]. Therefore, cellular antioxidant systems balance ROS production and oxidative stress. Superoxide dismutase 2 (Sod2) is solely found in the mitochondrial matrix and converts mitochondrial superoxide into H_2_O_2_ and O_2_ [55]. In the present study, we demonstrated that *B. burgdorferi* upregulates host *SOD2* gene expression in coinfected cells compared to *L. infantum* singularly infected cells. We confirmed *B. burgdorferi* exposure significantly decreased mtROS production in *L. infantum*-infected DH82 cells. Upon *B. burgdorferi* stimulation, human fibroblasts and PBMCs significantly upregulated *SOD2* gene expression [56,70]. Similar to our data, Wawrzeniak et al. demonstrated exposure to *B. burgdorferi* significantly reduced mtROS generation in a human neuroblastoma cell line [71]. Previous studies showed mtROS generation increases intracellular *Leishmania* killing in murine macrophages [44,45]. It has also been reported that mitochondria recruited to macrophage phagosomes via TLR1/2/4 signaling facilitate mtROS production and reduce bacterial burden [72,73]. Additionally, mtROS can reach pathogens within the phagosome via mitochondria-derived vesicles [74]. Even though these mechanisms remain undefined during anti-leishmanial defense [75], mtROS generation and its subsequent delivery to *L. infantum*-containing phagolysosome could be impaired during *B. burgdorferi* coinfection, contributing to parasitic survival. Interestingly, several studies have shown that enhanced mtROS generation promotes oxidation-induced cell death [76], another cellular mechanism inhibited by *B. burgdorferi* in our in vitro model. Therefore, by modulating matrix mtROS levels in *L. infantum*-infected cells, *B. burgdorferi* could potentially inhibit cell death and favor *L. infantum* survival in DH82 cells.

IFNγ promotes macrophage activation by inducing pro-inflammatory cytokine and chemokine production, upregulating antigen presentation and ROS production, sustaining antileishmanial responses and infection control [61,62]. At the same time IL-10 is produced as a negative feedback mechanism to limit excessive immunopathology [77]. CanL dog PBMCs exposed to *B. burgdorferi* produce high levels of IL-10 (Figure 5 and Figure 6). Our group has demonstrated that CD4 + T cells from dogs with polysymptomatic CanL show decreased IFNγ and increased IL-10 production in response to *Leishmania* antigen [78]. Consistent with our findings, van de Schoor et al. reported that *B. burgdorferi* minimally induces IFNγ production by human PBMCs, in addition to unchanged *IL-12p35* gene expression levels [79]. *B. burgdorferi* coinfection might redirect PBMC transcriptional programs away from IL-12 production and toward IL-23 production. Although by stimulating PBMCs with whole, live spirochetes, both our study and van de Schoor et al. likely missed other *B. burgdorferi*-specific responses, such as recognition of peptides and glycolipids by T cells, NK cells, and NKT cells [80,81]. Overall, exposure to live *B. burgdorferi* spirochetes induces pro-inflammatory responses but not remarkable Th1 cytokine production in PBMCs from *L. infantum*-seropositive dogs, supporting continued parasite maintenance.

In addition to *IL23p19*, we found significant upregulation of other Type 17 signature genes such as *IL6*, *IL1B*, *IL17A*, and *IL-22* in PBMCs from *L. infantum*-seropositive dog PBMCs after challenge with *B. burgdorferi*. We confirmed IL-17A production through ELISA. In addition, *B. burgdorferi* induces Th17 responses in mouse models of LD and in LD patients [82,83,84,85]. We did not observe modulation of *TGFB* expression, which contributes to Th17 cell differentiation in mice but has a less critical role in human Th17 differentiation [48], although we did not assess TGFβ protein or bioactivity changes. We also found increased IL-8, CXCL1, and CCL2 chemokines associated with neutrophilic and monocytic recruitment downstream of IL-17 signaling [48,65]. Regarding leishmaniasis, conflicting mouse and human studies associate IL-17A with both protection and pathology [86,87,88,89,90]. Canine studies show asymptomatic *L. infantum*-infection was associated with upregulated *IL17A* gene expression [50,51]. Sustained production of IL-17A leads to increased recruitment of neutrophils to inflammatory sites, which can contribute to deteriorating immunopathology [46]. Herein, we have demonstrated that *B. burgdorferi* exposure induces Type 17-like responses in PBMCs from *L. infantum*-seropositive dogs. Additional studies are required to confirm the development of IL-17A-producing Th17 cells, or other cellular sources of IL-17A, during *L. infantum* and tick-borne coinfections.

Concomitant infections are frequently observed in *Leishmania*-infected dogs, particularly bacterial (*B. burgdorferi*, *Ehrlichia* spp., *Anaplasma* spp.) infections [91]. *Ehrlichia* spp. has been shown to be the most frequent coinfection among dogs in major *L. infantum*-endemic areas such as Brazil and the Mediterranean basin [18,92,93]. *Ehrlichia* spp. immunity has been shown to also depend on development of CD4+ Th1 responses [94], which were impaired in CanL dogs. As *Ehrlichia* spp. reside intracellularly within monocytes and macrophages and subvert host cell immune responses through diverse evasion mechanisms [95], coexposure of *Ehrlichia* with *Leishmania* is likely to alter the same pathways and perhaps enhance survival of intracellular *Leishmania* parasites. Significant associations between exposure to *Ehrlichia* spp. and development of CanL have been previously described [18]. *Ehrlichia* spp. experimental conditions are extremely technically challenging for purification and quantification of bacterial load compared to *B. burgdorferi*, which do not require feeder cells for in vitro culture and do not form morulae [96,97]. Further efforts to overcome such challenges must occur for similar evaluation of *Ehrlichia* spp. and *L. infantum* coinfections.

Tickborne *B. burgdorferi* coinfection is common in dogs with CanL in the U.S. *B. burgdorferi* coinfection led to increased *L. infantum* burden within macrophages. We propose this is driven by enhanced viability of *L. infantum*-infected canine macrophages through *B. burgdorferi* inhibition of late apoptosis/necrosis. *B. burgdorferi* infection is not known to induce long-lived protective immunity [98]. Repeated spikes of systemic inflammation may occur after subsequent exposures, including Type 17 inflammation. These strong pro-inflammatory responses likely produce IL-10 regulation in CD4 + T cells, allowing parasites to replicate and contribute to CanL development. Increasing antigen loads in the context of sustained inflammation creates conditions where T cell exhaustion occurs, leading to disease progression [77]. Overall, evidence of tick-borne infections in dogs has been causally associated with CanL progression and serves as a sentinel of reported cases in humans not only in the U.S. [99,100,101] but in countries where vector-borne transmission of *Leishmania* occurs, like Brazil [102], India [103], the Middle East [104], and Mediterranean regions [20]. We propose reducing tick-borne diseases through available preventatives and interventions will make a significant difference in CanL control and prevent spread of *L. infantum* between dogs and humans.

## Figures and Tables

**Figure 1 pathogens-12-01128-f001:**
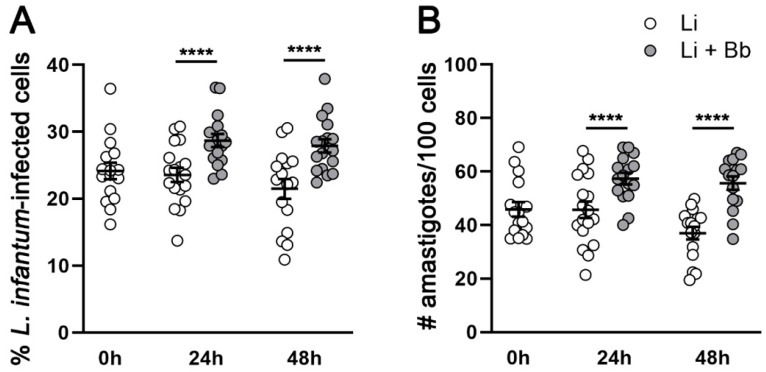
*B. burgdorferi* coinfection increases *L. infantum* infection rates and cellular burden. DH82 cells were exposed to *L. infantum* (1:10 MOI) for 24 h, and then exposed to live *B. burgdorferi* (1:25 MOI) for indicated time points. Cells were fixed with methanol and stained using HEMA 3 solutions. Frequencies of *L. infantum*-infected cells (**A**) and number of amastigotes (**B**) per 100 DH82 post-coinfections with live *B. burgdorferi* were assessed through light microscopy (100×). Blind and independent assessments were performed by two researchers. Data represent mean ± SEM of three independent experiments, performed in triplicates for each condition. *N* = 10–23 coverslips per group. Two-way ANOVA, Tukey’s multiple comparisons test. *p* **** < 0.0001. Li: *L. infantum* infection. Li + Bb: *L. infantum* and *B. burgdorferi* coinfection.

**Figure 2 pathogens-12-01128-f002:**
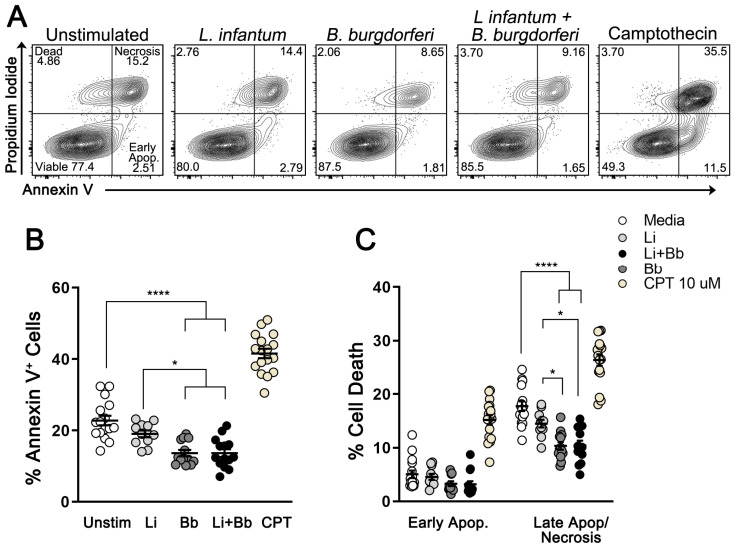
*B. burgdorferi* exposure decreases late apoptosis/necrosis rates in *L. infantum*-infected cells. Apoptosis assay using flow cytometry analysis of Annexin V and propidium iodide (PI) staining. DH82 cells were labeled with Annexin V (A488) and 1 µM PI at 24 h after no treatment; *L. infantum* infection (1:10 MOI); *B. burgdorferi* infection (1:25 MOI); coinfection, and 10 µM camptothecin (CPT; positive control). (**A**) Representative scatter plots of Annexin V (x-axis) vs. PI (y-axis). Bottom left quadrant: viable cells (Annexin V−/PI−); bottom right (Annexin V+/PI−): cells undergoing early apoptosis; top right (Annexin V+/PI+): cells that are in late-stage apoptosis/necrosis; top left (Annexin V−/PI+): dead cells. (**B**) Frequencies of DH82 cells undergoing cell death after infections and treatments. Ordinary one-way ANOVA, Tukey’s multiple comparisons test. *p* * = 0.0402 (Li vs. Bb); 0.0359 (Li vs. Li + Bb). *p* **** < 0.0001. (**C**) Frequencies of early apoptotic and late apoptotic/necrotic DH82 cells after infections and treatments. Two-way ANOVA, Sidak’s multiple comparisons test. Data are presented as mean ± SEM of triplicate experiments. *p* * = 0.01; *p* **** < 0.0001. Unstim: unstimulated (media); Li: *L. infantum* infection; Bb: *B. burgdorferi* infection; Li + Bb: *L. infantum* and *B. burgdorferi* coinfection. Apop.: apoptosis.

**Figure 3 pathogens-12-01128-f003:**
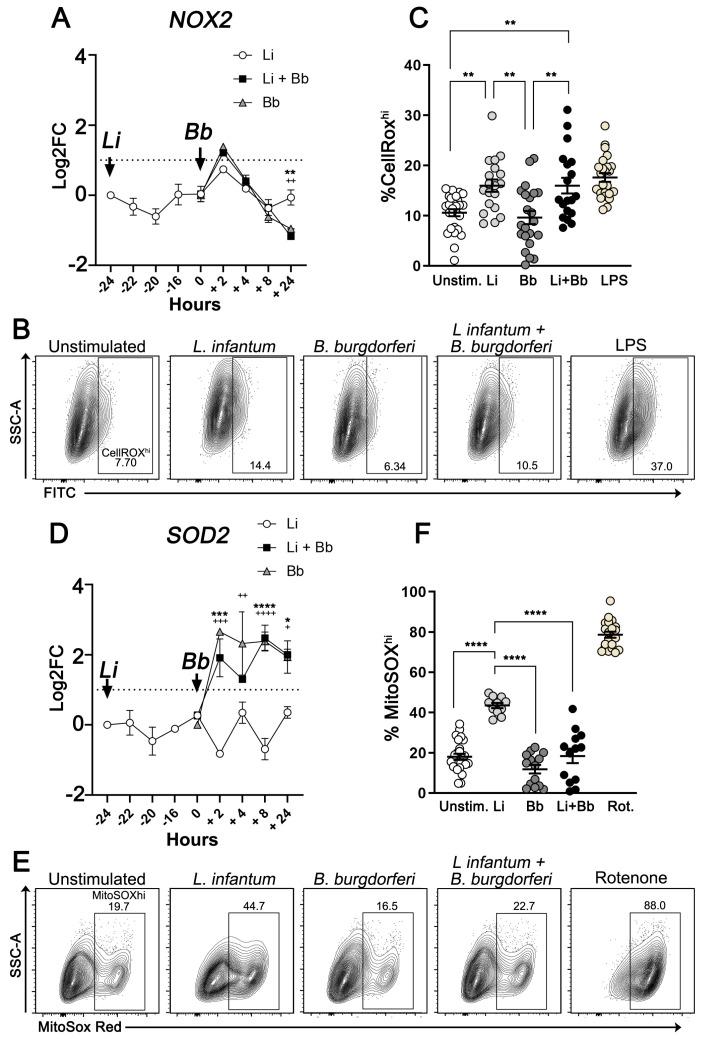
Altered production of reactive oxygen species (ROS) during *L. infantum* and *B. burgdorferi* coinfection in DH82 cells. Cells were exposed to *L. infantum* promastigotes and/or *B. burgdorferi* spirochetes. Gene expression of *NOX2* (**A**) and *SOD2* (**D**) were quantified via quantitative PCR and normalized using *ACTB* as an endogenous control. Data represent log transformed-fold change mean values relative to uninfected DH82 cells ± SEM of three independent experiments, with significance assessed via two-way ANOVA, Tukey’s multiple comparisons test. * *L. infantum* and *B. burgdorferi* coinfections vs. *L. infantum* infection. +*B. burgdorferi* infection vs. *L. infantum* infection. (**A**) *p* ** = 0.002, *p*
^++^ = 0.0014, (**D**) *p* * = 0.0114, *p*
^+^ = 0.0157, *p*
^++^ = 0.0022; *p* *** = 0.0006, *p*
^+++^ = 0.0002, *p* ****/^++++^ < 0.0001. Representative scatter plots showing percentages of CellROX^hi^ (**B**) and MitoSOX^hi^ (**E**) populations after exposure to medium, *L. infantum* promastigotes, *B. burgdorferi* spirochetes, and respective positive controls (LPS and rotenone). Percentage of CellROX^hi^ (**C**) and MitoSOX^hi^ (**F**) cells after *B. burgdorferi* exposure and indicated stimulations. Statistical significance was estimated using ordinary one-way ANOVA with post hoc Tukey test. (**C**) *p* ** = 0.0071 (unstimulated vs. Li/Li + Bb), 0.0016 (Bb vs. Li+Bb), (**F**) *p* **** < 0.0001, mean ± SEM of triplicate experiments.

**Figure 4 pathogens-12-01128-f004:**
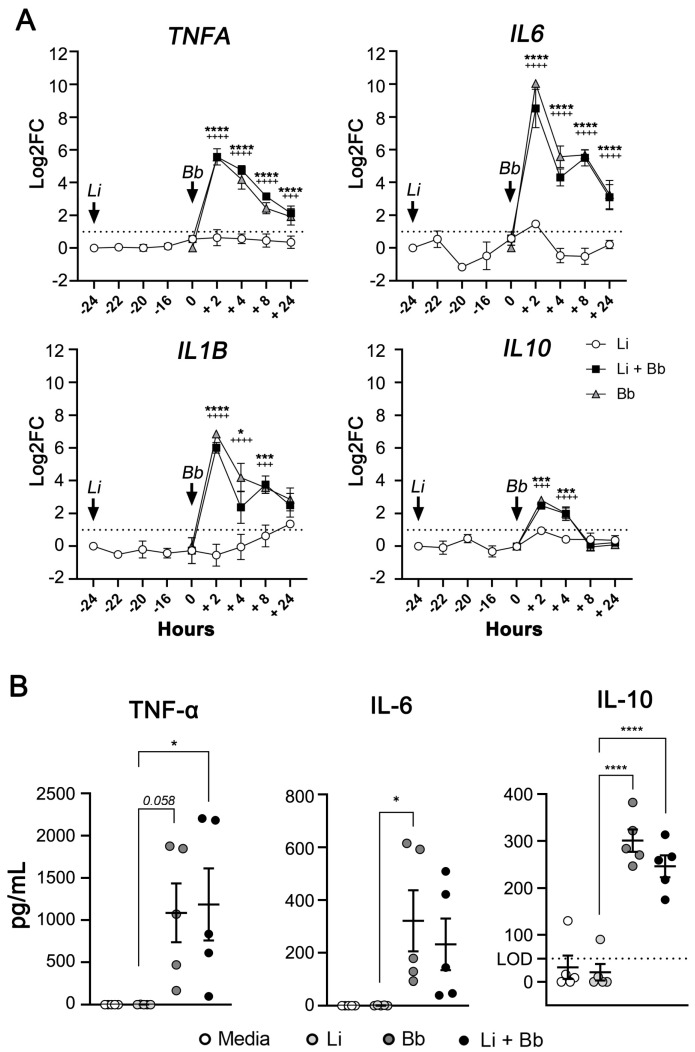
*B. burgdorferi* induces pro- and anti-inflammatory cytokines gene expression and production in *L. infantum*-infected DH82 cells. (**A**) Gene expression of *TNFA*, *IL6*, *IL1B*, and *IL10* were quantified via qPCR and normalized using *ACTB* as an endogenous control. Data represent log transformed-fold change mean values relative to uninfected DH82 cells ± SEM of three independent experiments using Two-way ANOVA, Tukey’s multiple comparisons test. * *L. infantum* and *B. burgdorferi* coinfection vs. *L. infantum* infection. +*B. burgdorferi* infection vs. *L. infantum* infection. *TNFA*, *p*
^+++^ = 0.0005; *p* ****/^++++^ < 0.0001; *IL6*, *p* ****/^++++^ < 0.0001; *IL1B*, *p* * = 0.0126; *p* *** = 0.0003; *p*
^+++^ = 0.0002; *p* ****/^++++^ < 0.0001); *IL10*, *p* *** = 0.0007 (2 h), 0.0002 (4 h); *p*
^+++^ = 0.0008; *p*
^++++^ < 0.0001). (**B**) Supernatants from DH82 cells were collected 24 h after exposure to *B. burgdorferi*. Release of TNF-α, IL-6, and IL-10 were quantified using Milliplex ELISA. Data represent mean values ± SEM of five independent experiments. Ordinary one-way ANOVA with Tukey post hoc test. TNF-α, *p* * = 0.0357; IL-6, *p* * = 0.0394; IL-10, *p* **** < 0.0001.

**Figure 5 pathogens-12-01128-f005:**
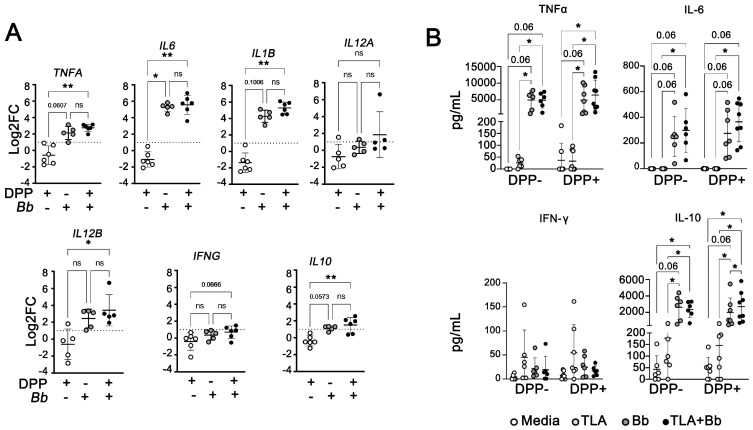
*B. burgdorferi* modulates gene expression and production of pro-inflammatory and regulatory cytokines in PBMCs from *L. infantum*-seronegative and seropositive dogs. PBMCs were stimulated with live *B. burgdorferi* spirochetes (1:10 MOI) for different timepoints. Cells were harvested at 24 h after infections for gene expression analyses. Supernatants were collected at 72 h after infections for ELISA. (**A**) Gene expression *TNFA*, *IL6*, *IL1B*, *IL12A*, *IL12B*, *IFNG*, and *IL10* were quantified by quantitative PCR and normalized using *ACTB* as an endogenous control. Data represent log transformed-fold change mean values (relative to unstimulated PBMCs from *L. infantum*-seronegative dogs) ±SEM of three independent experiments. Kruskal–Wallis with Dunn’s post hoc test. *TNFA*, *p* ** = 0.0027; *IL6*, *p* * = 0.0354; *p* ** = 0.005; *IL1B*, *p* ** = 0.0015; *IL12B*, *p* * < 0.05; *IL10*, *p* ** = 0.0089. *N* = 5–6 samples per group. (**B**) Release of TNF-α, IL-6, IFNγ, and IL-10 were quantified using Milliplex ELISA. Data represent mean values ± SEM of five independent experiments. Wilcoxon matched-pairs rank test, Holm–Sidak method. Adjusted *p* * = 0.03125. *N* = 5–8 samples per group.

**Figure 6 pathogens-12-01128-f006:**
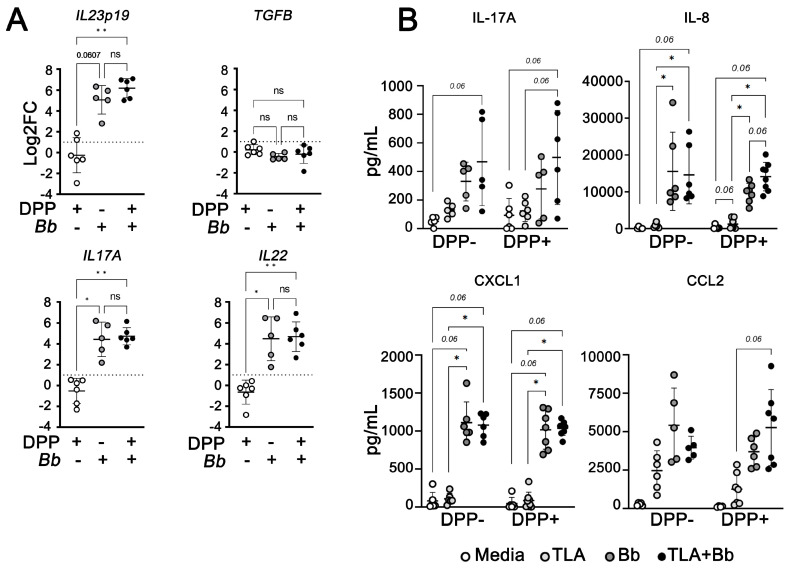
*B. burgdorferi* modulates gene expression and production of Type 17-related cytokines/chemokines gene expression in PBMCs from *L. infantum*-seronegative and seropositive dogs. (**A**) Gene expression of *IL23p19*, *TGFB*, *IL17A*, and *IL22* were quantified by quantitative PCR and normalized using *ACTB* as an endogenous control. Data represent log transformed-fold change mean values relative to unstimulated PBMCs from *L. infantum*-seronegative dogs ± SEM of three independent experiments. Kruskal–Wallis with Dunn’s post hoc test. *IL12p19*, *p* ** = 0.0027; *IL17A/IL22*, *p* * = 0.0199; *p* ** = 0.0089. (**B**) Release of IL-17A via sandwich ELISA, IL-8, CXCL1, and CCL2 via Milliplex ELISA. Data represent mean values ± SEM of five independent experiments. Wilcoxon matched-pairs rank test, Holm-Sidak method. Adjusted *p* * < 0.03125.

**Table 1 pathogens-12-01128-t001:** Age and sex distributions of *Leishmania*-seronegative and seropositive dogs.

	*Leishmania* Seroreactivity (DPP Test)	
	Negative (*N* = 6)	Positive (*N* = 8)
**Age**		
Mean (SD)	3.67 (1.63)	3.62 (1.06)
**Sex**		
Female	2 (33.3%)	3 (37.5%)
Male	4 (66.7%)	5 (62.5%)

SD: standard deviation.

## Data Availability

The data that support the findings of this study are available from the corresponding author upon reasonable request.

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
