# Peer review of "Modulation of Macrophage Redox and Apoptotic Processes to Leishmania infantum during Coinfection with the Tick-Borne Bacteria Borrelia burgdorferi"

_pathogens, 2023, doi:10.3390/pathogens12091128_

Round 1

Reviewer 1 Report

This study is well-designed, gives the readers new fundamental information, and can be accepted after some modification.

Figure 5 isn’t clarified, and the graphs' titles should be rearranged.

In Fig 5, A) Gene expression of TNFA, IL6, IL1B, IL12B, IL12A, B) IFNG, and C) IL10, the items B and C are unnecessary as all the cytokines analyzed by qPCR. Also, C is the release of cytokines by ELISA.

In Figure 6, A, and B is incorrect; please edit them based on the caption [A) Gene expression of IL23p19, TGFB, IL17A, and IL22 B) Release of IL-17A via sandwich ELISA, C) IL-8, CXCL1, and 457 CCL2 via Milliplex ELISA].

Please check all the figures and captions.

Please state the number of seropositive and seronegative samples used in the ELISA chapter in M&M (2.9) and the result section.

Author Response

This study is well-designed, gives the readers new fundamental information, and can be accepted after some modification.

We kindly thank the reviewer for their positive remarks on our work. All changes in the text of the manuscript are marked in red.

Figure 5 isn’t clarified, and the graphs' titles should be rearranged.

In the Figure 5, we preferred ordering the cytokines by their main role because it provided the strongest rationale. Although organizing the graphs according to their mention in the text could sound more efficient for readers, we believe it would not be as clear as arranging graphs by cytokine function. TNF-α, IL-6, IL-1β, and IL-12 are usually cytokines involved in activating T cells functions, while IFNγ and IL-10 are mostly T cell effector cytokines.

In Fig 5, A) Gene expression of TNFAIL6IL1BIL12BIL12A, B) IFNG, and C) IL10, the items B and C are unnecessary as all the cytokines analyzed by qPCR. Also, C is the release of cytokines by ELISA.

The reviewer is right. We edited the Figure 5 for including all cytokines analyzed by either qPCR or ELISA within their appropriate subsection (A and B, respectively; there is no more C). In addition, we swapped the IL12A and IL12B graphs to follow an alphabetical and logical order.

In Figure 6, A, and B is incorrect; please edit them based on the caption [A) Gene expression of IL23p19, TGFB, IL17A, and IL22 B) Release of IL-17A via sandwich ELISA, C) IL-8, CXCL1, and 457 CCL2 via Milliplex ELISA].

We combined all cytokines analyzed by qPCR into the same subsection (A). We also edited the figure legend according to the updated arrangement of the graphs (Figure 6A-B; there is no more C).

Please check all the figures and captions.

Done.

Please state the number of seropositive and seronegative samples used in the ELISA chapter in M&M (2.9) and the result section.

We agreed it was not clear how samples were grouped and analyzed by ELISA and qPCR. Indeed, PBMCs from three additional dogs were used for the ELISAs (N = 14). Therefore, we updated our demographics table (line 128). In addition, we added to the manuscript clarifications about types and numbers of different stimulation settings (lines 115, 148-150, 396, 408 and 410-411).

Reviewer 2 Report

General comments

The introduction is concrete, and the objective of the work is well justified. Overall, the study is well-designed and presented in a good way.

Note: It is highly recommended to review the entire document, as there are words that should be italicized.

Minor revisions

Introduction section

I recommended adding a graphical Abstract.

Line 64 - 66: Please add the corresponding reference.

Line 85 - 86: Please add the corresponding reference.

Materials and Methods section

In general, it is recommended to add and homogenize the brands and manufacturers of all reagents, as well as to add the country of origin.

2.1. Cell culture

Line 105: Please add the progeny of the macrophage cell lines. E.g. Canine Monocyte-macrophage (DH82), murine macrophage (RAW 264.7). (Check: PMID: 37299182)

Line 108: Fetal Bovine Serum (FBS) (Heat Inactivated)?? (Check: PMID: 37299182)

Line 111: CO2 should be exchanged for CO2.

Line 112: Please change Three days for 72 hours.

2.2. Animals

Line 122 should be: and canine Peripheral blood mononuclear cells (PBMC) isolation.

2.3. Parasites and bacteria

Line 134: Please change Roswell Park Memorial Institute (RPMI); add the brand of RPMI medium.

The medium was RPMI-1640?

2.4. PBMC stimulation assay

Do you have the protocol of the ethics committee for PBMC collection? Please add it to the supplementary material.

Why PBMC were not incubated in concave bottom plates, as these cells are not adherent? (Check PMID: 37109486)

Line 148: CO2 for CO2.

2.10. Statistical analyses

Please change Tukey’s post-test for Tukey’s post-hoc test,

Tukey’s or Sidak’s post-test for Tukey’s or Sidak’s post-hoc test.

Results section

In this section, I recommend improving the charts, some of them are too small, for example, in Figure 5 or 6.

Line 231-243: L. infantum, B. burgdorferi and in vitro must be in italics.

Line 245 – 253: L. infantum and B. burgdorferi must be in italics.

Line 311 – 356: There are many words that must be italicized.

References section

The scientific names must be in italics. Homogenize references according to the journal's guidelines.

Supplementary material

Please add as supplementary material the signed and endorsed protocol document of the ethics committee.

In Figure S1. B. burgdorferi and L. infantum should be in italics.

Author Response

The introduction is concrete, and the objective of the work is well justified. Overall, the study is well-designed and presented in a good way.

Note: It is highly recommended to review the entire document, as there are words that should be italicized.

We highly appreciated the reviewer’s favorable feedback on our work. All changes in the text of the manuscript are marked in red (except by the reference section). We were able to review the manuscript and reformat all the scientific names to italic.

Minor revisions

Introduction section

I recommended adding a graphical Abstract.

We thank the reviewer for their recommendation. We submitted a graphical abstract to be included in the publication.

Line 64 - 66: Please add the corresponding reference.

Done.

Line 85 - 86: Please add the corresponding reference.

Done.

Materials and Methods section

In general, it is recommended to add and homogenize the brands and manufacturers of all reagents, as well as to add the country of origin.

As requested by the reviewer, we homogenized the brands/manufacturers of all reagents mentioned in the Materials and Methods section. In addition, we included their respective countries of origins.

2.1. Cell culture

Line 105: Please add the progeny of the macrophage cell lines. E.g. Canine Monocyte-macrophage (DH82), murine macrophage (RAW 264.7). (Check: PMID: 37299182)

Done.

Line 108: Fetal Bovine Serum (FBS) (Heat Inactivated)?? (Check: PMID: 37299182)

Yes, we used heat inactivated FBS to supplement our media. This information as updated in the manuscript (line 109).

Line 111: CO2 should be exchanged for CO2.

Done.

Line 112: Please change Three days for 72 hours.

Done.

2.2. Animals

Line 122 should be: and canine Peripheral blood mononuclear cells (PBMC) isolation.

Done.

2.3. Parasites and bacteria

Line 134: Please change Roswell Park Memorial Institute (RPMI); add the brand of RPMI medium.

The medium was RPMI-1640?

We could not find any mention to “Roswell Park Memorial Institute” in the latest version of the manuscript provided for revision. To be doubly sure, we clarified we used RPMI-1640 as well as included the brand of the medium (line 135).

2.4. PBMC stimulation assay

Do you have the protocol of the ethics committee for PBMC collection? Please add it to the supplementary material.

Done.

Why PBMC were not incubated in concave bottom plates, as these cells are not adherent? (Check PMID: 37109486)

As described in the 2.4 section, we used 96-well round bottom plates (also known as concave bottom plates) for culturing PBMCs, as opposed to flat-bottom plates used for DH82 and RAW 267.4 cell cultures.

Line 148: CO2 for CO2.

Done.

2.10. Statistical analyses

Please change Tukey’s post-test for Tukey’s post-hoc test.

Done.

Tukey’s or Sidak’s post-test for Tukey’s or Sidak’s post-hoc test.

Done.

Results section

In this section, I recommend improving the charts, some of them are too small, for example, in Figure 5 or 6.

As requested by the reviewer, we improved the sizes of Figures 5 and 6.

Line 231-243: L. infantumB. burgdorferi and in vitro must be in italics.

Line 245 – 253: L. infantum and B. burgdorferi must be in italics.

Line 311 – 356: There are many words that must be italicized.

We were able to review the manuscript and reformat all the scientific names to italic.

References section

The scientific names must be in italics. Homogenize references according to the journal's guidelines.

Titles of all referenced articles were included as originally published, with or without italics per their respective journals.

We updated the format of our reference list to strictly abide by the journal’s guidelines.

Supplementary material

Please add as supplementary material the signed and endorsed protocol document of the ethics committee.

Done.

In Figure S1. B. burgdorferi and L. infantum should be in italics.

Done. Please, note that the abovementioned Figure is now the Figure S2 (changed after review).

Reviewer 3 Report

In the manuscript, entitled “Modulation of macrophage redox and apoptotic processes to Leishmania infantum during coinfection with tick-borne bacte-3 ria Borrelia burgdorferi”, by Pessoa-Pereira et al., submitted to Pathogens in August 2023 (id# pathogens-2557159), the authors explore the effect of co-infection with Borrelia burgdorferi and Leishmania in the context of in vitro infection, with mechanistic insight. The manuscript reads well and the data were well achieved. However, I think the interpretation of the data must be re-considered for the sake of accuracy Bellow, I list my reservations in a point-by-point manner.

Major issues:

1.      My main issue is related with the overall message. The authors highlight the co-infection condition as something unique which is not the case. The phenotype of co-infection is similar to that in the frame of infection with Borrelia alone. This means that Leishmania is taking advantage of the modifications caused by Borrelia infection. Not there co-infection is associated with some kind of an additive effect.

2.      Also related with the above point, I do think you need to double-check your statistical analysis and tone down the statements that claim differences between cells infected with just Leishmania and those co-infected with Borrelia and Leishmania. This refers particularly to the data in figures 5 (e.g. significance in the difference of IL-10 levels in DPP+ samples stimulated with Bb or TLA+Bb is hardly believable), and figure 6 (e.g. “We observed higher 442 levels of IL-17A production from PBMCs exposed to both B. burgdorferi and TLA than by 443 cells stimulated with either pathogen alone [Figure 6C]” – no significance).

3.      There is no explanation for the MOIs selected. Please clarify your decision – is the experimental setting relevant considering natural infection? - and discuss whether you would expect different results in the context of higher infection burdens.

Minor points:

1.      The data are quite interesting. Taking this in consideration you can extend your discussion. Are tick-infected dogs more likely to develop asymptomatic infection (as opposed to the self-healing scenario?). Are asymptomatic dogs more likely to develop active disease if the get infected with borrelia?

2.      Please ensure that all Latin species names are italicized (e.g. Line 232).

3.      The results in supplementary figure 1 are not a complete parallel. You clearly see parasite elimination in this experimental context. I would consider writing that into the text – e.g. co-infection diminishes parasite elimination.

4.      In line 370 you mention that Leishmania infection led to an increase in proinflammatory cytokine expression which is not true (values below the dashed line, which I am not sure what it denotes).

Author Response

In the manuscript, entitled “Modulation of macrophage redox and apoptotic processes to Leishmania infantum during coinfection with tick-borne bacte-ria Borrelia burgdorferi”, by Pessoa-Pereira et al., submitted to Pathogens in August 2023 (id# pathogens-2557159), the authors explore the effect of co-infection with Borrelia burgdorferi and Leishmania in the context of in vitro infection, with mechanistic insight. The manuscript reads well and the data were well achieved. However, I think the interpretation of the data must be re-considered for the sake of accuracy. Bellow, I list my reservations in a point-by-point manner.

We thank the reviewer for their feedback and insightful comments. All changes in the text of the manuscript are marked in red.

Major issues:

1. My main issue is related with the overall message. The authors highlight the co-infection condition as something unique which is not the case. The phenotype of co-infection is similar to that in the frame of infection with Borrelia This means that Leishmaniais taking advantage of the modifications caused by Borrelia infection. Not there co-infection is associated with some kind of an additive effect.

The reviewer is right about the similar phenotypes presented by cells exposed only to B. burgdorferi and to both pathogens. We agree B. burgdorferi coinfections favored Leishmania survival within macrophages, which is in line with our manuscript conclusions. However, we believe the condition (coinfection) can be considered unique for the specific outcomes provoked in L. infantum-infected dogs. The clinical presentation and immunopathology of canine Lyme Disease is different from canine leishmaniosis. Although B. burgdorferi infections in Leishmania-seronegative dogs are highly inflammatory, most of the dogs will remain clinically normal and only present specific signs (Lyme arthritis and nephropathy) when left untreated and the spirochetes have already disseminated (PMID: 20207198). In contrast, L. infantum-infected dogs may initially develop an asymptomatic infection, but it eventually escalates to a progressive, chronic visceral disease within months to years. Multiple factors have been associated with disease progression, such as breed, immunocompetence, and comorbid diseases. We previously demonstrated that, in an event of a tick-borne coinfection, the risk of canine leishmaniosis progression and mortality increases significantly (PMID: 30674329). Therefore, based on our epidemiological data and the findings of our manuscript, B. burgdorferi coinfection can be considered an additive effect as dogs die more frequently and earlier than dogs not exposed as well as it increases parasite burden in host cells.  

2. Also related with the above point, I do think you need to double-check your statistical analysis and tone down the statements that claim differences between cells infected with just Leishmaniaand those co-infected with Borrelia and Leishmania. This refers particularly to the data in figures 5 (e.g. significance in the difference of IL-10 levels in DPP+ samples stimulated with Bb or TLA+Bb is hardly believable), and figure 6 (e.g. “We observed higher 442 levels of IL-17A production from PBMCs exposed to both  burgdorferi and TLA than by 443 cells stimulated with either pathogen alone [Figure 6C]” – no significance).

In Figure 5B (IL-10 ELISA), there was a significant statistical difference between PBMCs exposed to Bb and to TLA+Bb (p = 0.031). We used the Wilcoxon matched-pairs signed-rank test. Although it might show similar means, the groups differed in the sum of signed ranks (W) (please see attached table). The critical value of W for 7 paired samples is 4 considering α = 0.05 for one sided test (2 if two-sided). As W value (-28) was less than the critical value (4), we can reject the null hypothesis and consider the difference between groups statistically different.

We agree there is a significant overlap of values between those groups – which one would say they are actually very similar. However, if we subset the paired PBMCs from DPP+ dogs that were stimulated with either Bb or TLA+Bb, we can see that PBMCs consistently produce more IL-10 in response to TLA+Bb than Bb alone (please see attached figure).

We agree that the differential production of IL-17A did not reach statistical significance between the stimulated groups for DPP- and DPP+ dogs, but there was a clear increasing trend in PBMCs exposed to Borrelia (p = 0.06). We believe we could have achieved statistical significance if we had a larger sample size. Nevertheless, as requested by the reviewer, we updated our statement appropriately (line 447-448). 

3. There is no explanation for the MOIs selected. Please clarify your decision – is the experimental setting relevant considering natural infection? and discuss whether you would expect different results in the context of higher infection burdens.

We understand that, although providing relevant insights into the pathogenesis of infectious diseases, experimental MOIs do not fully emulate the complexity of natural infections. However, a MOI of 10 Leishmania parasites and Borrelia spirochetes per cell is commonly used in in vitro and ex vivo infection studies (i.e., PMID: 35352952; 27624408; 27391120; 34000990). The MOI of 1:25 or similar for B. burgdorferi infections has also been used in ex vivo studies (i.e., PMID: 19909078, 12946326) mainly to achieve optimal peak of infection efficiency and immune responses. In our in vitro experimental setting, preliminary data revealed no statistical significance when comparing different MOIs on DH82 cell viabilities (please see attached figure). Therefore, we would not expect that higher burdens of B. burgdorferi would impact cell death rates. Although several studies have demonstrated that increasing MOIs (i. e., 1:50; 1: 100) induce higher production of cytokines and chemokines (i.e., PMID: 28655322, 22232682), those supraphysiological numbers would hardly reflect the systemic loads found in dogs exposed to Borrelia.

Minor points:

1. The data are quite interesting. Taking this in consideration you can extend your discussion. Are tick-infected dogs more likely to develop asymptomatic infection (as opposed to the self-healing scenario?). Are asymptomatic dogs more likely to develop active disease if the get infected with borrelia?

We thank the reviewer for appreciating our data.

It is still not fully known whether a dog will either resolve infection, remain asymptomatic, or progress towards a clinical disease. However, a self-healing phenotype is a less common event in our population of US hunting foxhounds when compared to pet dogs. We briefly described in our Introduction section that tick-borne pathogen exposures, including to Borrelia, are strongly associated with canine leishmaniosis progression and mortality in our US cohort of L. infantum-infected foxhounds (PMID: 30674329). Our present results support our previous epidemiological findings by demonstrating that B. burgdorferi favors L. infantum survival as well as reduces mitochondrial ROS generation and cell death – indicating impaired infection control by L. infantum-infected macrophages. In addition, we confirmed that B. burgdorferi infections are highly inflammatory, and thus could contribute to canine leishmaniosis pathology development. We described how strong pro-inflammatory responses would affect L. infantum infection control and, consequently, disease progression in the last paragraph of the discussion (lines 576-591). We explored this idea in more detail on a side prospective study from our group (prepared for submission), where we captured immune events occurring before and after tick-borne coinfection exposures in naturally L. infantum-exposed dogs without clinical disease.

2. Please ensure that all Latin species names are italicized (e.g. Line 232).

We were able to review the manuscript and reformat all the scientific names to italic.

3. The results in supplementary figure 1 are not a complete parallel. You clearly see parasite elimination in this experimental context. I would consider writing that into the text – e.g. co-infection diminishes parasite elimination.

The reviewer is correct when describing how the murine RAW 264.7 macrophages eliminate parasites more efficiently than DH82 cells. However, although at different extents, L. infantum-infection rates and parasite burden also significantly increased in RAW 264.7 macrophages. For better clarifications, we updated our manuscript for describing the differences in parasite control between both cell lines as requested by the reviewer (lines 245-246).

4. In line 370 you mention that Leishmania infection led to an increase in proinflammatory cytokine expression which is not true (values below the dashed line, which I am not sure what it denotes).

We thank the reviewer for bringing it to our attention. We mistakenly wrote down that L. infantum-infected macrophages increased the transcription of the cytokines analyzed in our work – which was a typo. We updated the sentence to clarify that L. infantum does not increase pro-inflammatory response in DH82 macrophages (line 375). 

Reviewer 4 Report

The studies present experiments involving co-infections with Leishmania infantum and Borrelia burgdorferi which are relevant to clinical outcomes of canine leishmaniasis in regions that are co-endemic for tick-borne pathogens.   Effects of the co-infections on diminishing the cell death and generation of mitochondrial ROS by a canine macrophage cell line were described.  The main consequence is that co-infections with B. burgdorferi results in increased intracellular parasite infection levels and burdens within DH82 macrophages.  Ex vivo co-stimulation of PBMCs from dogs seronegative or seropositive for L. infantum infection was also performed, with the results showing that B. burgdorferiinduces a limiting IFNg response, but cytokines and chemokines associated with Th17 differentiation.  Overall, while the in vitro and ex vivo readouts are well removed understanding how co-infections involving these pathogens actually influence the clinical outcome of canine leishmaniasis, they offer some useful insights as to how B. burgdorferi might promote the chronicity and pathology associated with leishmaniasis.  

Some specific concerns should be addressed: 

The authors argue that the L. infantum infection increase 24-48 hr after B. burgdorferi exposure was due mainly to better survival of the co-infected cells.  However, based on the data in fig. 2, the effect of B. burgdorferi exposure on protecting cells from apoptosis was the same on cells in the uninfected cultures as compared to the cells in the L. infantum infected cultures.  Wouldn’t the infected and uninfected cells within the same B. burgdorferi treated culture be equally protected against apoptosis?  If so, why would their relative frequencies change?  

To conclude that the parasite burdens within infected cells was also promoted by exposure to B. burgdorferi, the number of amastigotes per infected cell at each time point should also be shown.  Carrying out the infections for a longer time (72-96 h) would also be useful to show that the intracellular survival/growth is affected by B. burgdorferi.   

line 370: “As expected, L. infantum-infected macrophages did increase inflammatory cytokine transcription compared to uninfected cells (Figure 4A).”   Is this meant to read “did not”?

line 410:  “PBMCs from L. infantum-seropositive dogs showed a slightly stronger gene induction after B. burgdorferi exposure, with significantly increased TNFAIL6IL1BIL10  and IL12B (<0.01) compared to unstimulated L. infantum-seropositive dog PBMCs.”   This sentence is unclear.   There seems to be no significant difference in the transcript levels or secreted cytokine levels in B. burgdorferi-stimulated PBMCs from seropositive compared to seronegative dogs.  

Line 471: "Leishmania spp. and B. burgdorferi both can suppress apoptosis of infected cells as an evasion mechanism [30,34, 65], so we hypothesized that the two pathogens may synergistically decrease apoptosis, consequently favoring L. infantum maintenance within coinfected cells.” It should be mentioned that no synergistic effects on suppressing apoptosis were observed.

Title is a bit misleading as it conveys that the experiments were actually carried out using tick-borne bacteria.   The bacteria used were no more tick-borne than the parasites used were sandfly borne.  

Author Response

The studies present experiments involving co-infections with Leishmania infantum and Borrelia burgdorferi which are relevant to clinical outcomes of canine leishmaniasis in regions that are co-endemic for tick-borne pathogens.   Effects of the co-infections on diminishing the cell death and generation of mitochondrial ROS by a canine macrophage cell line were described.  The main consequence is that co-infections with B. burgdorferi results in increased intracellular parasite infection levels and burdens within DH82 macrophages.  Ex vivo co-stimulation of PBMCs from dogs seronegative or seropositive for L. infantum infection was also performed, with the results showing that B. burgdorferi induces a limiting IFNg response, but cytokines and chemokines associated with Th17 differentiation.  Overall, while the in vitro and ex vivo readouts are well removed understanding how co-infections involving these pathogens actually influence the clinical outcome of canine leishmaniasis, they offer some useful insights as to how B. burgdorferi might promote the chronicity and pathology associated with leishmaniasis.  

We greatly thank the reviewer for their valuable comments and feedback. All changes in the text of the manuscript are marked in red.

Some specific concerns should be addressed: 

The authors argue that the L. infantum infection increase 24-48 hr after B. burgdorferi exposure was due mainly to better survival of the co-infected cells.  However, based on the data in fig. 2, the effect of B. burgdorferi exposure on protecting cells from apoptosis was the same on cells in the uninfected cultures as compared to the cells in the L. infantum infected cultures. Wouldn’t the infected and uninfected cells within the same B. burgdorferi treated culture be equally protected against apoptosis? If so, why would their relative frequencies change?

To clarify, we proposed that the enhanced L. infantum survival after Borrelia exposure could be caused by the decreased cell death (lines 485-488, 577-579), but further studies are necessary to determine a causal association between the events (lines 488-491).

Although not impacting early apoptotic events, the decreased cell death rates observed after B. burgdorferi exposures were primarily associated with reduced late apoptosis (Figure 2C) – which we believe it could contribute to the intracellular survival of parasites.

Most likely, within the same Borrelia-exposed cell culture, there will be pools of uninfected and infected cells. However, we do not believe that these cells would be equally protected from cell death for Borrelia spirochetes would have to either interact with or infect cells to promote the observed effect. Thus, because we found different rates of cell death between uninfected and Borrelia-exposed cell cultures, we assume that the frequencies of cells directly exposed to spirochetes were greater than those not exposed within the same culture. We acknowledge, however, that more specific assays would be necessary to precisely determine infection frequencies within each well (i. e., flow cytometry, immunofluorescence).

To conclude that the parasite burdens within infected cells was also promoted by exposure to B. burgdorferi, the number of amastigotes per infected cell at each time point should also be shown. Carrying out the infections for a longer time (72-96 h) would also be useful to show that the intracellular survival/growth is affected by B. burgdorferi.  

To determine the parasite burdens, we instead showed the numbers of amastigotes per 100 cells at each time point, which is a method frequently used in many in vitro and ex vivo Leishmania studies (i. e., PMID: 31856207; 10688819).

We agree that allowing coinfections for longer time points could provide more insights on the dynamics of the parasite infection control. However, in our experimental setting, DH82 macrophages would achieve confluency at late time points (72-96h) and thus undergo stress and cell death – which would compromise our findings.

line 370: “As expected, L. infantum-infected macrophages did increase inflammatory cytokine transcription compared to uninfected cells (Figure 4A).”   Is this meant to read “did not”?

The reviewer is correct, and we thank them for bringing it to our attention. We mistakenly wrote down that L. infantum-infected macrophages increased the transcription of the cytokines analyzed in our work – which was a typo. We updated the sentence to clarify that L. infantum does not increase pro-inflammatory response in DH82 macrophages (line 375). 

line 410: “PBMCs from L. infantum-seropositive dogs showed a slightly stronger gene induction after B. burgdorferi exposure, with significantly increased TNFAIL6IL1BIL10 and IL12B (<0.01) compared to unstimulated L. infantum-seropositive dog PBMCs.”   This sentence is unclear.   There seems to be no significant difference in the transcript levels or secreted cytokine levels in B. burgdorferi-stimulated PBMCs from seropositive compared to seronegative dogs. 

The reviewer is correct when pointing out to no significant differences between PBMCs from seropositive and seronegative dogs exposed to Borrelia spirochetes. However, in the abovementioned sentence, we intended to highlight that, because the response was slightly higher than in PBMCs from seronegative dogs, the transcripts were significantly upregulated when compared to unstimulated PBMCs. We updated the sentence for better clarification (lines 413-419).

Line 471: "Leishmania spp. and B. burgdorferi both can suppress apoptosis of infected cells as an evasion mechanism [30,34, 65], so we hypothesized that the two pathogens may synergistically decrease apoptosis, consequently favoring L. infantum maintenance within coinfected cells.” It should be mentioned that no synergistic effects on suppressing apoptosis were observed.

As the reviewer requested, we updated the text by mentioning the event was not synergistic (lines 479-480; 488).

Title is a bit misleading as it conveys that the experiments were actually carried out using tick-borne bacteria. The bacteria used were no more tick-borne than the parasites used were sandfly borne.

The reviewer is correct that neither infected ticks nor sand flies had been used in our experiments. However, we believe our title does not imply that we applied a vector transmission in our model, but simply that Borrelia is a tick-borne pathogen. One of the implications of our work is that preventing tick-borne diseases with available preventatives would have a significant impact in CanL control and prevent transmission to humans. Therefore, we prefer to keep the “tick-borne” mention in the title.

Reviewer 5 Report

The manuscript entitled " Modulation of macrophage redox and apoptotic processes to Leishmania infantum during coinfection with tick-borne bacteria Borrelia burgdorferi” developed an in vitro co-infection model of L. infantum and B. burgdorferi using a canine macrophage cell line, DH82 cells, to assess the impact of these co-infections. The authors demonstrated how B. burgdorferi alters the macrophage phenotype, favouring Leishmania replication. In fact, the two parasites synergistically could decrease apoptosis, consequently favouring the maintenance of L. infantum in cells. Moreover, this work supports, with appropriate in vitro models and using both molecular and serological methods, how effective tick prevention and thus management of the risk of co-infection can be critical strategies to control the progression of L. infantum in dogs. I have read, with interest this paper which is overall well-written, well-structured and clear, and should be of great interest to the readers. I suggest some changes to be made:

--write “L. infantum” and “B. burgdorferi” in italic;

-“in vitro” should be written in italic.

-Line 153- add bibliography

-Line 163- add bibliography

With these appropriate modifications, the manuscript could be published, because it could provoke a debate in the scientific world.  Finally, once these changes are made, I recommend that the article be accepted for publication in this journal.

Author Response

The manuscript entitled " Modulation of macrophage redox and apoptotic processes to Leishmania infantum during coinfection with tick-borne bacteria Borrelia burgdorferi” developed an in vitro co-infection model of L. infantum and B. burgdorferi using a canine macrophage cell line, DH82 cells, to assess the impact of these co-infections. The authors demonstrated how B. burgdorferi alters the macrophage phenotype, favouring Leishmania replication. In fact, the two parasites synergistically could decrease apoptosis, consequently favouring the maintenance of L. infantum in cells. Moreover, this work supports, with appropriate in vitro models and using both molecular and serological methods, how effective tick prevention and thus management of the risk of co-infection can be critical strategies to control the progression of L. infantum in dogs. I have read, with interest this paper which is overall well-written, well-structured and clear, and should be of great interest to the readers. I suggest some changes to be made:

We thank the reviewer for their positive comments on our work.

--write “L. infantum” and “B. burgdorferi” in italic;

-“in vitro” should be written in italic.

We were able to review the manuscript and reformat all the scientific names to italic.

-Line 153- add bibliography

We checked the latest version of the manuscript provided for revision, and, according to it, the reviewer requested adding references for culturing DH82 and RAW 267.4 cell lines into coverslips. We added a reference for the method involving seeding cells into coverslips for analyzing infection rates (line 155).

-Line 163- add bibliography

We checked the latest version of the manuscript provided for revision, and, according to it, the reviewer requested adding references for the Annexin V-PI apoptosis assay. We strictly followed the manufacturer`s protocol, as described in line 170.

With these appropriate modifications, the manuscript could be published, because it could provoke a debate in the scientific world.  Finally, once these changes are made, I recommend that the article be accepted for publication in this journal.

We greatly appreciate the reviewer’s considerations.

Round 2

Reviewer 3 Report

Although no major changes were made in the manuscript, I understand the authors' points. Please consider some notes:

1. I was under the impression you added 0.1 parasites per cell. If you added 10 parasites/bacteria per cell, the multiplicity of infection should be 10:1.

2. With respect to the statistical analysis with respect to IL-10, if you have paired samples, please connect them with lines so the reader can appreciate the reproducible increase in Bb versus Bb + L. infantum.

3. With regards to the new sentence on the contrast between murine and canine macrophages (lines 244-246), it does not reflect the accuracy of the data. You are implying that you have lower parasite elimination in comparison with the observed with canine cells which is not the case.

Author Response

1. I was under the impression you added 0.1 parasites per cell. If you added 10 parasites/bacteria per cell, the multiplicity of infection should be 10:1.

We thank very much the reviewer for bringing it to our attention. Our MOIs were based on cell to parasite/bacteria ratios. After reviewing the literature (i.e., PMID: 34832536; 32277055; 31629910; 18490754), we found MOI ratios can be reported in either order (cell:pathogen or pathogen:cell) as long as it is clarified in the methods. Therefore, we added the clarification about the MOI ratios in our Methods section (line 135).

2. With respect to the statistical analysis with respect to IL-10, if you have paired samples, please connect them with lines so the reader can appreciate the reproducible increase in Bb versus Bb + L. infantum.

We understand individual dogs from the same group (either DPP- or DPP+) can present different responses after different stimulation settings. However, even though the observations (samples) were paired, our conclusions were not drawn from checking individual changes, but from evaluating the differences at the population level. Therefore, we believe changing the graph to a connected scatter plot is not necessary to visualize the information.  

3. With regards to the new sentence on the contrast between murine and canine macrophages (lines 244-246), it does not reflect the accuracy of the data. You are implying that you have lower parasite elimination in comparison with the observed with canine cells which is not the case.

We thank the reviewer for their insight. We re-ran our statistical analyses to include comparisons between initial infection (0 h time point) and post-infection time points (24 and 48 h). We then re-evaluated our findings and, for better clarification and data accuracy, changed their description as follows (line 237-252):

We found that DH82 macrophages maintained a steady parasite infection rate and burden over 48 hours in culture [Figure 1]. However, B. burgdorferi coinfected macrophages showed significantly higher Leishmania infection frequencies compared to L. infantum-infected cells at both 24 h (21.8% increase; p = 0.0001) and 48 h (31.4% increase; p = 0.0001) post B. burgdorferi addition [Figure 1A]. Numbers of intracellular parasites were also significantly higher in B. burgdorferi co-infected macrophages than L. infantum-infected cells at 24 h (25.6% increase; p = 0.0001) and 48 h (68.1% increase; p <0.0001) post B. burgdorferi addition [Figure 1B]. We additionally measured the difference in parasite infection and burden rates between Leishmania infection alone or B. burgdorferi coinfection in RAW 264.7 murine macrophage cell line [Figure S2]. RAW 264.7 macrophages showed a signification reduction in parasite infection and burden at 24 h and 48 h compared to the initial infection (p <0.0001). However, there was not a significant reduction in parasite infection and burden at 24 h or 48 h compared to the initial infection among B. burgdorferi coinfected RAW 264.7 cells. Further, at 24 h and 48 h time points, the coinfected group showed a significantly higher rate of Leishmania infection and burden compared to the Leishmania-infected macrophages alone”.

Please, note we updated the p values from the previous comparisons as we included the initial infection time point within the analyses. 

Reviewer 4 Report

Reviewer comments adequately addressed.

Author Response

Reviewer comments adequately addressed.

We immensely thank the reviewer for their insightful comments that led to possible improvements in the current version.